# Full and Hybrid Multiscale Lubrication Modeling

Arthur Francisco [1,*,†] and Noël Brunetière [2,†]

1    Institut Pprime, CNRS, Université de Poitiers, ISAE-ENSMA, 16021 Angoulême, France
2    Institut Pprime, CNRS, Université de Poitiers, ISAE-ENSMA, 86962 Futuroscope Chasseneuil, France
*    Correspondence: arthur.francisco@univ-poitiers.fr; Tel.: +33-545-673-249
†    These authors contributed equally to this work.

**Abstract:** The numerical solution for the lubrication of parallel rough surfaces cannot be obtained using the well-known flow factors of Patir and Cheng. Nor can it be determined using homogenization techniques. Is there an alternative, besides a purely long-term deterministic way of solving the problem? The present paper aims at proposing a multiscale approach in order to reduce the computing time, specific to deterministic resolutions, while maintaining good accuracy. The configuration is a parallel rough surface slider, with imposed hydrodynamic operating conditions. The domain consists of independent macro-elements, on which the Reynolds equation is solved. Then, the macro-element boundaries are adjusted to ensure global mass conservation. In its hybrid version, the algorithm replaces some well-chosen macro-elements by simple linear finite elements. The results clearly show the potential of our method. Because the lubrication of each macro-element can be processed independently, the multicore architecture of the processor is exploited. Even if the performance depends on the ratio roughness/height, the computing time is half than for the classical deterministic method, with a few percent errors. The work concludes with some recommendations on the configurations for which the multiscale method is best suited, such as surfaces with short correlation lengths.

**Keywords:** lubrication; multiscale modeling; rough surfaces

## 1. Introduction

In a lubricated contact, when the fluid enters a converging zone, the pressure increases and reaches a maximum near the minimum of the film thickness. Assuming that a diverging area follows the converging entrance, the fluid is stretched and the pressure decreases. The Reynolds equation as originally expressed does not account for the cavitation area in a realistic way and abundant literature has followed. Among the methods that guarantee the Reynolds conditions, the penalty method and the well-known Elrod's algorithm are quite easy to implement. The penalty method promoted by Wu [1] treats both the film area and the cavitation area by introducing a penalty term into the Reynolds equation. When the pressure falls below the cavitation pressure $P_c$, its value is forced to $P_c$. Whilst providing an accurate pressure boundary, the method fails in predicting the film reformation boundary. The seminal Elrod and Adams algorithm [2,3] splits the Reynolds equation by the means of a switch function to ensure the equation validity in both the gaseous and liquid areas. Mass conserving is guaranteed in both areas; however, the location of the boundaries is mesh dependent.

Let us describe the cavitation phenomenon a little more. If the pressure falls below the ambient pressure, three situations can be considered [4]: (a) the dissolved gas is released and forms bubbles, (b) the bubbles already present expand, and (c) the fluid, which contains no gas, evaporates. The cases are referred to as a (a) gaseous cavitation, (b) pseudocavitation, and (c) vaporous cavitation. In a more recent study, Bai et al. [5] propose a review on the cavitation in a thin liquid layer, in which a section is dedicated to the hydrodynamic cavitation. However, the authors do not distinguish the different cavitation phenomena as reported by Braun and Hannon [4]. The simplified taxonomy (a)-(b)-(c) is that the

latter present does not include adsorption effects, as studied by Belova et al. [6], on the heterogeneous cavitation. Although rough surfaces are likely to adsorb much more gas than smooth surfaces because of their fractal nature, the related phenomenon significantly complicates the lubrication modeling of rough surfaces and therefore is not addressed in the present work.

Regarding the gaseous cavitation in oil, Li et al. [7] and Song and Gu [8] consider the lubricant as a mixture of pure liquid and air, as the result of the dissolution and the release of air in an instantaneous equilibrium saturation state. The model satisfies the classical Jakobson–Floberg–Olsson (JFO) conditions, but it does not account for the gaseous cavitation rate, as performed by Hao and Gu [9] and more recently by Ding et al. [10].

Ransegnola et al. [11] model both the vaporous and gaseous cavitation in an oil-lubricated bearing, predicting not only the cavitation area but also the distribution of the gas, vapor, and liquid states. However, with respect to water as a lubricant, Magaletti et al. [12] provide a graph that shows the cavitation pressure of ultra-pure water as a function of temperature. The remarkable data are the cavitation pressure of the water at ambient conditions, about $-120$ MPa, because of its high tensile stress. This very low value makes one think that behind the common cavitation in water, there is essentially a gaseous cavitation or even a pseudocavitation.

In the present work, the fluid is assumed to be a biphasic mixture: air bubbles are present in an incompressible liquid, which leads to the pseudocavitation in the diverging zones. Considering the mixture as a homogeneous media allows for the use of the Reynolds equation in the whole contact zone: the fluid rheology is modified in the depressurized zones and is constant in the full-film zones. Three advantages are brought with this model presented by Brunetière [13] and referred to as the "Lubricant General Model" (LGM). (1) It is a handy model because once the fluid density and the fluid viscosity are defined, the Reynolds equation is solved in the same way, whatever the zone. (2) The cavitation area transition is smooth, making it particularly suitable for multiscale meshes. (3) The varying density model is a mass conserving one; as such, it accurately accounts for the film rupture and reformation.

Grützmacher et al. [14] propose a review of the multiscale approaches about texturing in tribology. The classification that is proposed is well suited to distinguish the different strategies related to the lubrication of rough contacts. In particular, two classes are of importance here: the analytical multiscale methods and the numerical multiscale modeling.

Lubricated contacts may require multiscale approaches for various reasons. Heavily loaded contacts can lead to high gradient pressures, such as in the outlet spike in point contacts. The problem can be addressed with fine meshes, but without speedup convergence algorithms, such as multigrid methods, the computing time becomes prohibitive. The multiscale approach is then a means to accelerate the convergence of the Reynolds equation resolution. The mesh can also exhibit different element scales based on the pressure gradient: the steeper the pressure field, the finer the mesh. In both situations, the Reynolds equation is not modified. The above common techniques belong to the numerical multiscale modeling class.

The analytical multiscale methods involve treatments applied to the Reynolds equation. When roughness influences the fluid flow because of the small film thicknesses, instead of a deterministic resolution, the equation can be enriched with flow factor modifiers [15,16]. The flow factors are determined with a few of the statistical properties of the rough surface, which limit the accuracy of the results. However, once the flow factors are determined, the surfaces being considered smooth, coarse meshes can be used, decreasing the computing effort. This stochastic approach has since been improved to cope with the micro-cavitation to handle a broader variety of surface roughnesses [17], but it remains global: the local effects of the roughness on the pressure cannot be captured. In order to get rid of the stochastic approximation of the roughness, the Reynolds equation can be viewed as a set of equations solving different wavelength pressure problems [18–20]. Indeed, an asymptotic expansion of the pressure is written with respect to a scalar related to the roughness

wavelength, leading to a modified Reynolds equation. The process, based on rigorous mathematical developments, is called homogenization and is compatible with any periodic roughness. Homogenization techniques have received more attention in recent years than flow factor modifiers with the works of, among others, de Boer et al. [21,22], de Boer and Almqvist [23], and Han et al. [24]. In Rom et al. [25], the reader can find the homogenization advantages that explain its greater development. However, flow factor modifiers seem to be more widely used to date, such as in [26–30].

Computing flow factors and homogenization suffer from a common drawback: for flat parallel rough surfaces, no pressure build-up exists, although it is experimentally observed [31–34]. A compromise between the deterministic and the stochastic methods has been proposed by Brunetière and Wang [35]. The Reynolds equation is filtered: above an $f_c$ roughness frequency, averaging is used; and below $f_c$, a deterministic solution is computed on a coarse grid. For this analytical multiscale method, the computational effort is much less than for the deterministic case, but the micro-cavitation is not taken into account. Pei et al. [36,37] use a Guyan reduction to condensate finite element cells. In doing so, the general linear system is smaller than the one obtained with the usual finite element method. A further reduction is obtained while defining the master/slave nodes on the cell boundary. Even if the bandwidth is larger, the computing time is up to five times smaller than for the conventional finite element method (FEM). The major drawback of this numerical multiscale modeling is, however, that the cavitation is not taken into account.

The present paper, belonging to the numerical multiscale approaches, is a step beyond the work of Brunetière and Francisco [38]: the domain is divided into macro-cells inside which a deterministic FEM is used to solve the Reynolds equation; then, the macro-cells are linked together using a mass-conserving principle to cover the whole domain. In addition, when the average film thickness over a macro-element is sufficient to ignore the roughness, the macro-element is not submeshed but rather replaced by a linear finite element: it is the hybrid version of the algorithm.

## 2. Materials and Methods

A flat lubricated rough contact is modeled with the functioning parameters presented in Table 1.

**Table 1.** Contact characteristics.

| Parameter | Value |
|---|---|
| Minimum of the film thickness | 0.05 to 10.0 µm |
| Fluid type | mixture (liquid and gas) |
| Top-scale mesh, $Ne_x \times Ne_y$ elements | $128 \times 128$ |
| Bottom-scale meshes, $ne_x \times ne_y$ elements | $8 \times 8$ |
| Sliding speed along $x$ | $5.0\,\mathrm{m\,s^{-1}}$ |
| Fluid viscosity | $10^{-3}\,\mathrm{Pa\,s}$ |
| Fluid density | $10^3\,\mathrm{kg\,m^{-3}}$ |
| Ambient pressure | $10^5\,\mathrm{Pa}$ |

The rough surface used in the simulation is a numerically generated $1025 \times 1025$ points bi-Gaussian surface, which is typical of surfaces used in tribological applications, Figure 1. It is half the one used by Brunetière and Francisco [38]. The surface characteristics are presented in Table 2.

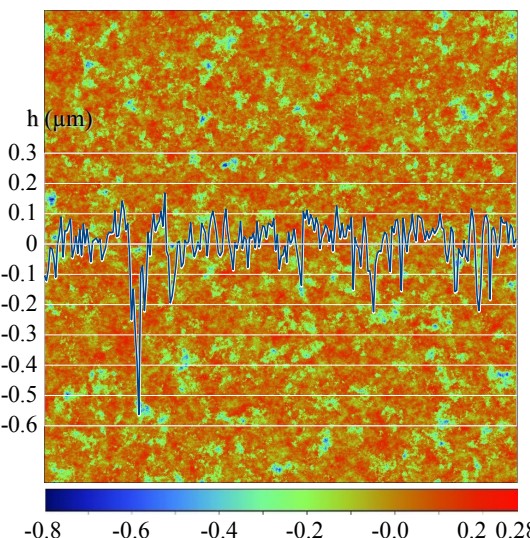

**Figure 1.** Bi-Gaussian rough surface used for the numerical simulations.

**Table 2.** Rough surface characteristics.

| Parameter | Value |
|---|---|
| Surface type | numerically generated |
| Numerical size $n \times n$ points (nodes) | $1025 \times 1025$ |
| Physical size, $L_x \times L_y$ | $1\,\text{cm} \times 1\,\text{cm}$ |
| RMS roughness, $Sq$ | $0.1\,\mu\text{m}$ |
| Roughness skewness, $SSk$ | $-1.4$ |
| Roughness kurtosis, $SKu$ | $6.3$ |
| Autocorrelation length, $Sal(s = 0.2)$ | $124\,\mu\text{m}$ |
| Ratio of domain length $L_x$ to correlation length $Sal$, $Rcl$ | $80.4$ |

### 2.1. Model Equations

Under the usual lubrication assumptions, the Reynolds equation writes:

$$\nabla \left( \frac{\rho h^3}{\mu} \boldsymbol{\nabla} p \right) = \nabla (6\boldsymbol{u}\rho h) \tag{1}$$

where the left-hand side is the diffusive part of the equation, and the right-hand side is the convective part if $\rho$ is pressure-dependent.

As fully detailed in the supplementary material (SM), Section S1, the Bubnov-Galerkin method leads to spurious oscillations if the convective part becomes greater than the diffusive part. As a solution, whereas the Poiseuille term is weighted by the shape functions $N_i$, the Couette term is weighted by upwind functions $\tilde{N}_i$ defined as:

$$\tilde{N}_i = N_i + \frac{\alpha}{2} l_e \frac{\partial N_i}{\partial x} \tag{2}$$

where

- $l_e$ is the streamline length of an element;
- $\alpha = \coth(P_e) - \dfrac{1}{P_e}$, $P_e$ is the Peclet number of an element;
- $P_e = 3\dfrac{\mu u(\partial \rho / \partial p)}{\rho h^2} l_e$.

It is the Petrov–Galerkin scheme for which different functions than the shape functions are used for the weak formulation. Here, the shape and the weighting functions are chosen linear.

Hence, after summation over the entire domain $\Omega$, one obtains the weak form of the Equation (1) for each internal node $i$:

$$-\iint_{\Omega_i} \frac{\rho h^3}{\mu} \boldsymbol{\nabla} N_i \cdot \boldsymbol{\nabla} p \, d\Omega + \iint_{\Omega_i} \overline{\rho h} \, \boldsymbol{u} \cdot \boldsymbol{\nabla} N_i \, d\Omega = 0 \tag{3}$$

where

- $\Omega_i$ is the subdomain where $N_i$ is not null;
- $\overline{\rho h} = (\rho h)_k \bar{N}_k$;
- $\bar{N}_k = N_k - \dfrac{\alpha}{2} l_e \dfrac{\partial N_k}{\partial x}$.

As explained in the SM, Subsection S2.1 and S2.2.

During the resolution process, the left part of Equation (3) is actually not null but equal to a residual $R_i$. The Newton–Raphson method leads to the determination of the increment $\delta p_j$ at the node $j$, such that $R_i(p + \delta p_j) = 0$, hence:

$$R_i + \frac{\partial R_i}{\partial p_j} \delta p_j = 0 \tag{4}$$

Thus, for all $\Omega$ internal nodes, the pressure increments are iteratively determined after the resolution of the system $[K]\{\delta p\} = -\{R\}$, detailed in SM, Subection S2.3.

The implementation of the lubricant general model is straightforward because it is a mixture of an incompressible liquid—subscript "l"—and a gas—subscript "g". Noting $r_g$, the specific gas constant, $\lambda$ the gas mass fraction, and $T$ the ambient temperature:

- For an incompressible liquid:

$$\rho = \rho_l$$
$$\frac{\partial \rho}{\partial p} = 0$$

- For a perfect gas:

$$\rho = \frac{p}{r_g T}$$
$$\frac{\partial \rho}{\partial p} = \frac{1}{r_g T}$$

- For a mixture of both:

$$\rho = \frac{1}{\frac{1-\lambda}{\rho_l} + \frac{\lambda r_g T}{p}}$$
$$\frac{\partial \rho}{\partial p} = \frac{\lambda \rho_l^2 r_g T}{\left(p(1-\lambda) + \lambda r_g T \rho_l\right)^2}$$

## 2.2. Tested Cavitation Algorithms

As mentioned earlier, in the present work, a varying density model is used for the fluid—a mixture of water and air bubbles—ensuring mass conserving throughout the contact. Even if this pseudocavitation model proves to be accurate [13], it is nevertheless compared to the penalty method and to the Elrod algorithm: the former, for its ease of use, and the latter, for its recognized accuracy.

### 2.2.1. Elrod's Algorithm, the Reference

The full-film area and the cavitation area ($p = P_c$) are explicitly separated: in the former, the Reynolds equation applies, whereas in the latter, a pure Couette flow is modeled with the lubricant fraction $\theta$: $\nabla(6\rho\theta u h) = 0$. Both areas are handled in a single modified Reynolds equation thanks to a switch function that de/activates the Poiseuille contribution.

For each node $i$, if $p_i > P_c$, then $\theta_i = 1$; otherwise, if $p_i = P_c$, then $\theta_i < 1$. It is undoubtedly the most used algorithm because it guarantees the JFO conditions, ensuring mass conserving everywhere inside $\Omega$, and no parameter needs to be tuned. Elrod's algorithm is therefore used here as the reference to compare the penalty method and the general model.

### 2.2.2. The Penalty Method

A means to force sub-cavitation pressure to remain at the cavitation pressure $P_c$ consists of adding a penalty term in the equation to solve. The penalty correction must act as a source term whenever the pressures fall below $P_c$ and it must vanish whenever the pressures are above $P_c$. Thus, modifying the Equation (1), one obtains:

$$\nabla\left(\frac{\rho h^3}{\mu}\boldsymbol{\nabla}p\right) = \nabla(6\boldsymbol{u}\rho h) + \kappa(p - P_c)^- \tag{5}$$

where $(p - P_c)^-$ is the $(p - P_c)$ negative part and $\kappa$ an arbitrary high coefficient that makes negligible the other terms. If $p > P_c$, then $(p - P_c)^- = 0$; otherwise, $(p - P_c)^- \simeq 0$.

For each finite element $e$, a local linear system is written in the form $[K_e]\{\delta p\} = \{B_e\}$ and $\kappa$ is set to $\tau \times \max\{B_e\}$ and $\tau = 10^k$. For a fair comparison of the penalty method to the Elrod algorithm, the parameter $\tau$ must be well chosen.

To do so, the values $k \in [\![2, 8]\!]$ are tested on a rough parabolic slider, Figure 2, and the results are compared to Elrod's results. The parabolic slider highlights the pressure build-up ending with a large cavitated zone; the multi-lobed slider highlights the film reformations; and the flat slider is the challenging configuration for which only the deterministic model gives satisfactory results, so far.

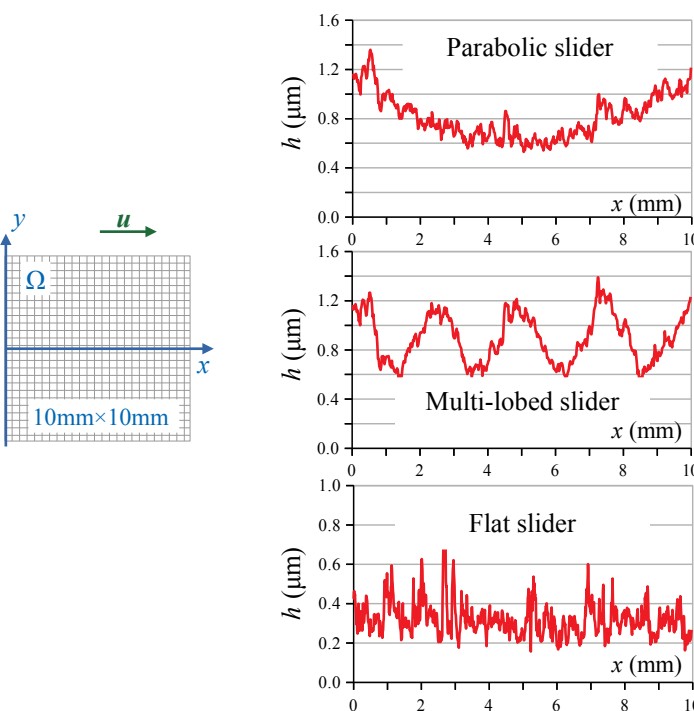

**Figure 2.** Lubricated contact domain $\Omega$, $(x, y) \in [0, 10] \times [-5, 5]$, and slider profiles at $y = 0$.

Figure 3 shows the differences between the penalty method pressures and JFO pressures for the parabolic slider (blue curve, scale on the right). Several orders of magnitude of $\tau$ parameter are represented, yet no difference between the curves is observed in the full-film zone. However, as expected, tiny differences are located in the cavitated area, zoomed in on Figure 4.

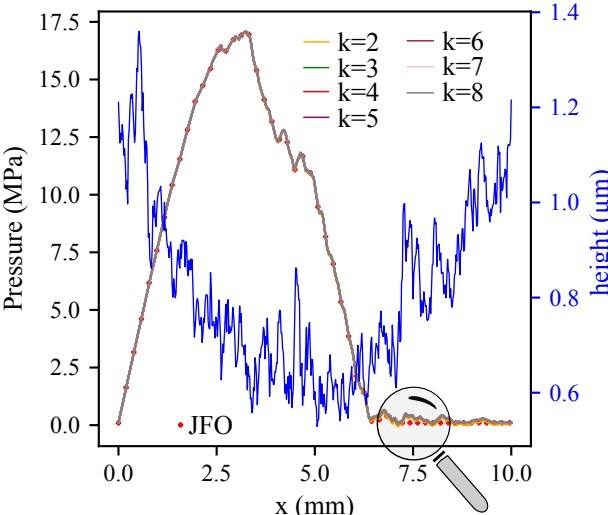

**Figure 3.** Influence of $\tau = 10^k$ on the pressure for the parabolic slider at $y = 0$. The film thickness is the blue curve whose scale is shown on the right.

If closer attention is paid to the cavitated area, Figure 4, the penalty results become indistinguishable for $k > 4$.

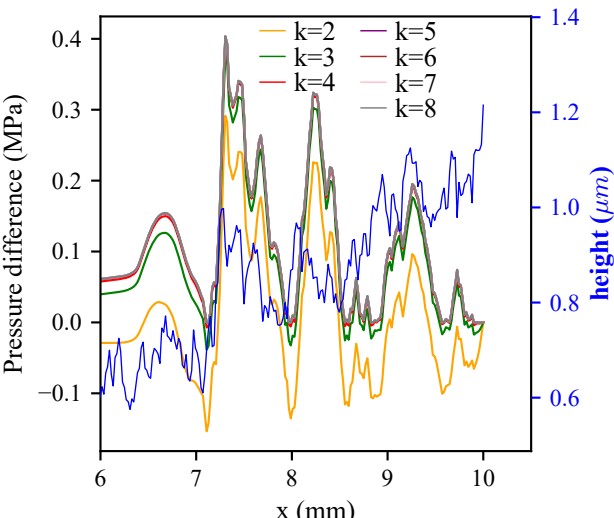

**Figure 4.** Influence of $\tau = 10^k$ on the pressure for the parabolic slider at $y = 0$. Zoom around $x = 8$ mm.

As suggests Table 3, the computing time increases with $k$, and values above 5 bring no additional benefit. Therefore, $k = 5$ is chosen to test the method further.

It is to be noted that no additional variable is required, unlike Elrod's algorithm, and the implementation is quite easy: it surely explains why many authors choose the penalty method for Hertzian lubricated contact.

**Table 3.** Influence of $\tau$ on the results. $n_{it}$ stands for the number of Newton–Raphson iterations until convergence, and *cpu* is the computing time.

|  | JFO | $k=2$ | $k=3$ | $k=4$ | $k=5$ | $k=6$ | $k=7$ | $k=8$ |
|---|---|---|---|---|---|---|---|---|
| $n_{it}$ | 33 | 29 | 34 | 39 | 43 | 47 | 52 | 63 |
| *cpu* (s) | 317.7 | 266.7 | 309.5 | 361.9 | 402.2 | 434.7 | 606.3 | 614.1 |
| Load (N) | 480.7 | 481.7 | 486.2 | 487.3 | 487.5 | 487.5 | 487.6 | 487.6 |

When it comes to film reformation, to what extent is the penalty method not advantageous because of its non-mass-conserving character? Figure 5 answers the question by highlighting the large differences between both methods on a multi-lobed rough surface. The penalty method does not suit our needs, namely a rough flat slider with scattered film reformations.

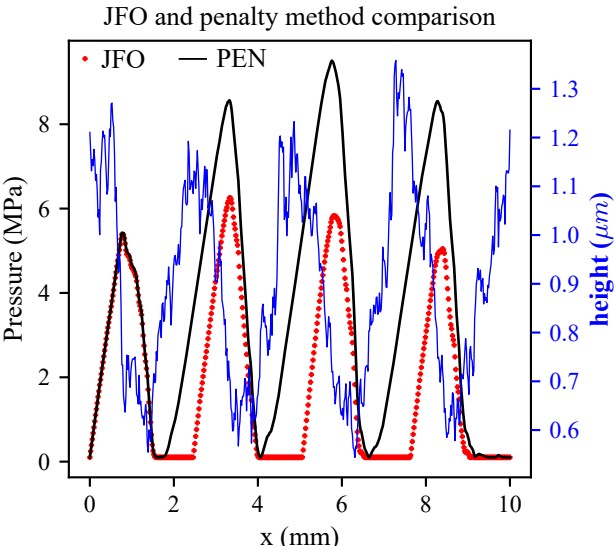

**Figure 5.** Comparison of pressure profiles obtained in JFO conditions (JFO) and with the penalty method (PEN), with a multi-lobed slider at $y = 0$.

### 2.2.3. The Lubricant General Model

The LGM model—also referred to as MIX, as in 'MIXture', in the present paper—lays on the assumption that the lubricant contains a constant mass fraction $\lambda$ of gas. It is a rather simple model inspired by physical considerations. When the lubricant is stretched, the gas expansion makes the lubricant volume increase. Following Braun and Hannon classification [4], the LGM belongs to the pseudocavitation models.

One way of choosing a suitable value for the parameter $\lambda$ is to fit numerical results on experimental data, as proposed by Brunetière [13] with $m = 7.5$ in the relationship $\lambda = m \times 10^{-5}$, $10^{-5}$ being the common order of magnitude.

Another way of determining $m$ is to consider Henry's law that states that *the amount of dissolved gas in a liquid is proportional to its partial pressure above the liquid. The proportionality factor is called Henry's law constant* [39].

Then, $C_a = H_a^{cp} p$, where $C_a$ (mol m$^{-3}$) is the concentration of a species in the liquid phase, $p$ (Pa) is the partial pressure of that species in the gas phase under equilibrium conditions, and $H_a^{cp}$ is Henry's constant.

Considering the amount of dissolved oxygen and nitrogen in water, we can write:

$$\begin{cases} C_{O_2} = H_{O_2}^{cp} P_{atm} & \text{with} & H_{O_2}^{cp} = 1.3 \times 10^{-5}\,\text{mol m}^{-3}\,\text{Pa}^{-1} \\ C_{N_2} = H_{N_2}^{cp} P_{atm} & \text{with} & H_{N_2}^{cp} = 6.5 \times 10^{-6}\,\text{mol m}^{-3}\,\text{Pa}^{-1} \end{cases}$$

Because partial pressures add, the mass in kilogram of dissolved air per liter of water is $\lambda'$:

$$\lambda' = (0.21 \times C_{O_2} \times 32 + 0.79 \times C_{N_2} \times 28) \times 10^{-6}$$
$$= 2.3 \times 10^{-5} \, \text{kg L}^{-1}$$

where for oxygen—molar mass $32 \, \text{g mol}^{-1}$, atmospheric pressure fraction 21%—the mass dissolved per liter of water is $0.21 \times C_{O_2} \times 32 \times 10^{-6}$. Finally, the mass fraction $\lambda$ is $\lambda'\rho$, which yields the same value.

$H_a^{cp}$ values are taken from the huge database provided by Sander [40]. The result is of the same order as the previous one ($7.5 \times 10^{-5}$). It can however be objected that dissolved gas implies gaseous cavitation, which is not the assumption for the LGM. However, according to Grando et al. [41], *absorption usually occurs at a much slower rate than release, and the liquid may not be able to absorb the gas in the flow time available during the positive pressure region.* Even if the latter statement relates to a journal bearing, we suppose it to be applicable here. The scenario proposed is hence that once the dissolved gas is released in a cavitation area, the lubricant remains a two-phase liquid.

To figure out the importance of $\lambda$, a parametric study is carried out to quantify its effect on the numerical results.

Figures 6 and 7 show the pressure profile along $y = 0$ for different values of $m$, with $\lambda = m \times 10^{-5}$ and JFO's model. Because the global behavior is the same, the detailed view highlights the subatmospheric pressures in a cavitated zone. High values of $m$ mean more dissolved gas in the lubricant, so the volume increases sooner and the pressure decreases at a slower rate. As a consequence, with increasing values of $\lambda$, the pressure has higher values below the atmospheric pressure in the cavitated zones. It is worth noting that the cavitation pressure chosen for the JFO model is the atmospheric pressure, which is higher than observed values. However, compared to the full-film pressures, the impact remains limited. The same applies to the resulting load, for which the cavitated areas account for a little.

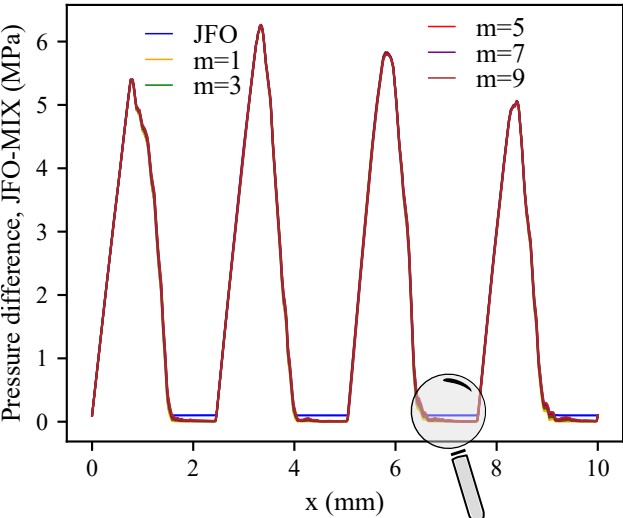

**Figure 6.** General model (MIX) pressure with different values of $m$ in $\lambda = m \times 10^{-5}$ for the multi-lobed slider at $y = 0$.

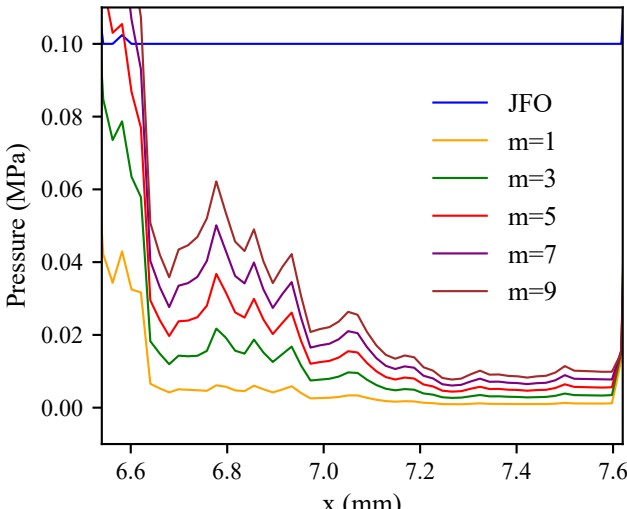

**Figure 7.** Zoom on a cavitated zone of the multi-lobed slider at $y = 0$—the general model (MIX) pressure is plotted with different values of $m$ from $\lambda = m \times 10^{-5}$ and compared to Elrod's algorithm.

$\lambda$ is chosen so that it provides a small computing time while keeping the normal load $N$ close to JFO's; as Table 4 suggests, $\lambda = 5.0 \times 10^{-5}$, which is rather close to $7.5 \times 10^{-5}$.

**Table 4.** Influence of $\lambda$ on the results. $n_{it}$ stands for the number of Newton–Raphson iterations until convergence, and *cpu* is the computing time.

|          | JFO   | $m = 1$ | $m = 3$ | $m = 5$ | $m = 7$ | $m = 9$ |
|----------|-------|---------|---------|---------|---------|---------|
| $n_{it}$ | 22    | 19      | 17      | 16      | 18      | 22      |
| *cpu* (s)| 231.9 | 181.4   | 161.6   | 152.2   | 171.1   | 226.8   |
| Load (N) | 127.9 | 125.7   | 127.1   | 127.8   | 128.0   | 127.8   |

To conclude this chapter, the LGM pseudocavitation model, once tuned, is chosen here, as it offers accuracy and efficiency close to those of a classical JFO model. Moreover, the cavitation boundaries are mesh-independent, which gives it a definite advantage.

### 2.3. The Multiscale Approach

The multiscale approach introduced by Brunetière and Francisco [38] is detailed hereafter. Let us consider a domain $\Omega$ as illustrated in Figure 8. $\Omega$, the hatched zone, is discretized according to $n_e^t \times n_e^t$ elements mesh. The superscript $^t$ refers here to "Top Scale" (TS). Each TS element is called a macro-element because it is also discretized, thus providing a "Bottom Scale" (BS) $n_e^b \times n_e^b$ elements mesh. At the BS level, the whole $\Omega$ mesh is $(n_e^t \cdot n_e^b) \times (n_e^t \cdot n_e^b)$ elements: it is the fine mesh on which deterministic simulations are carried out. In the present case, it is a square regular mesh, but the method can apply to any configurations.

To help the understanding, a variable value is located on the grid with two indices (row, column). Once the pressures $P_{IJ}$ are initialized at the TS level, a TS element, "D", for instance, in Figure 8a, is considered. The parameters are the four nodal pressures $\{P_{32}, P_{42}, P_{43}, P_{33}\}$, interpolated on the macro-element boundary $\Gamma_D$. The variables $p_{ij}$, the inner pressures, are the unknowns determined with the Reynolds equation resolution, Figure 8d.

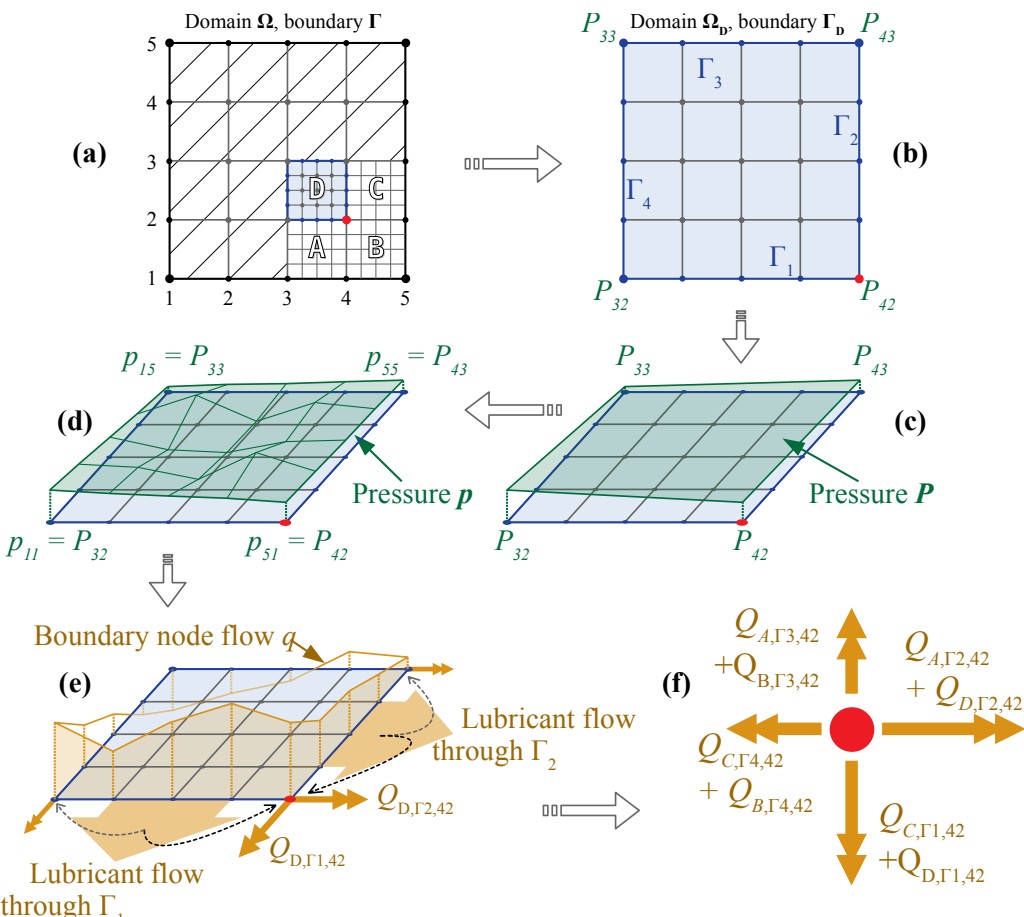

**Figure 8.** Overview of the multiscale approach. (**a**) The top-scale (TS) mesh is a $4 \times 4$ linear quadrangle mesh. Each TS element (macro-element) is subdivided to form a bottom-scale (BS) mesh, e.g., macro-elements A–D. The TS pressure is interpolated on BS elements (**b**,**c**), and then, the Reynolds equation is solved to determine the BS pressure (**d**). (**e**) The mass flow is calculated on the boundary $\Gamma_1 \cup \Gamma_2 \cup \Gamma_3 \cup \Gamma_4$, and it is distributed on the corner nodes. Once the macro-elements A, B, C, D are treated, the mass flow balance can be performed on the red node (**f**).

Each macro-element can be treated independently, which allows for multithreaded computations. When the whole set of macro-element computations is ended, the macro-element boundary mass flows are calculated, Figure 8e, and nodal mass flows are distributed on the macro-element node corners. Thereafter, for each TS node $IJ$, the residual nodal mass flow $Q_{IJ}$ is calculated. At the TS level, the pressures are considered correct if $Q_{IJ} \simeq 0$.

One of the key ideas of the multiscale iterative scheme is to perturb the TS nodal pressures to assess the effects on the TS nodal mass flows, Figure 9a,b. Hence, properly modifying the nodal pressures makes the TS nodal mass flow increase: it is a classical Newton–Raphson procedure where the derivatives of the mass flow balance with respect to the pressure must be determined. Here, numerical derivatives are used with $\delta P = P/100$.

When the system $[\partial Q/\partial P]\{\delta P\} = \{Q\}$ is determined and solved, the TS pressures are updated, and the process starts a new iteration until a stop criterion is met. Because all macro-elements are treated in the same way, the multiscale approach is called full multiscale (FMS).

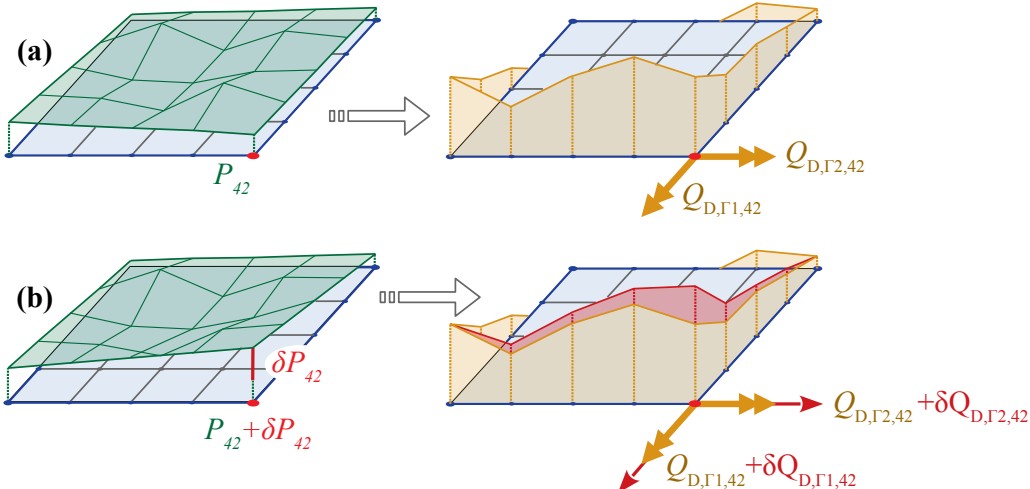

**Figure 9.** How the pressure affects the mass flow at the macro-element corner. (**a**) The mass flow is determined on the macro-element boundary and distributed on the corners. (**b**) A small variation is introduced in the corner pressure and the mass flows are updated. When all macro-elements are treated, the TS matrix $[\partial Q/\partial P]$ can be assembled.

For better multiscale time computing performances and accuracy, Brunetière and Francisco [38] recommend two macro-elements per unit *Rcl* as a rule of thumb. As 1024 is a power of 2, the nearest values are $Ne_x = Ne_y = 128$ and $Ne_x = Ne_y = 256$. As the results do not differ much for both configurations, the chosen grid is $Ne_x = Ne_y = 128$.

The results of the deterministic method—classical FEM resolution of the Reynolds equation—are used as the reference. Two accuracy criteria are chosen: the normal load $N$ as a global criterion and the maximum of the film pressure $P_{max}$ as a local criterion. Two computing times are recorded: the total computing time—sum over all threads—$T_t$ and the apparent computing time $T_a$. The computing performance is assessed dividing the computing times by the computing time of the deterministic simulation. The same goes for the computing accuracy which is obtained dividing the normal $N$ load and the maximum of the film pressure $P_{max}$ by the deterministic related values.

## 3. Results and Discussion

### 3.1. Deterministic Case

The solver MUMPS v5.4.1—MUltifrontal Massively Parallel Solver [42,43]—is used to iteratively solve the linear systems $\{R\} + [\partial R/\partial P]\{\delta P\} = 0$. As expected, the computing time required for the deterministic model decreases when the minimum of the film thickness $h_{min}$ increases; see Figure 10. It is worth noting that the computing time decreases almost linearly (in a log–log scale) until there is no more carrying capacity, $N = 10.0\,\text{N}$—the atmospheric pressure loading. From a numerical point of view, it also appears that the ratio of the total computing time to the apparent computing time is a constant around 3. It is as if three threads were simultaneously used during the whole case resolution. Actually, there are eight available threads, but the numerical treatments do not fully use them, as shown in Figure 11.

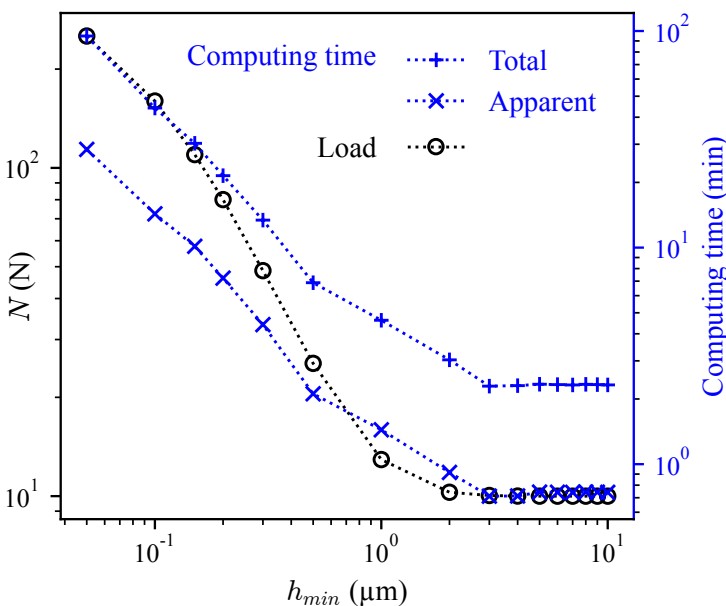

**Figure 10.** Minimum of the film thickness $h_{min}$ influence on the computing time of the deterministic model. The left scale is used for the load carrying capacity $N$, and the right scale is used for the computing time (total $T_t$ and apparent $T_a$).

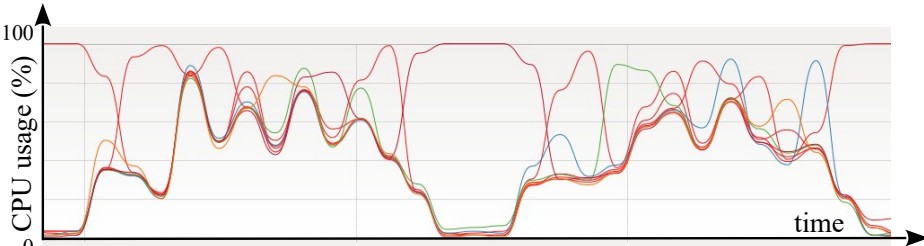

**Figure 11.** CPUs usage of MUMPS when solving a deterministic case. The lines represent the different threads.

### 3.2. FMS, Full Multiscale

In the FMS configuration, the whole set of the macro-elements is subdivided. It is reminded that there are $128 \times 128$ macro-elements; each macro-element is subdivided into $8 \times 8$ elements, for an overall number of $1025 \times 1025$ nodes.

Figure 12 clearly shows that, first, the errors on $N$ and $P_{max}$ do not exceed 6%. The computing time gain is roughly between 20 and 70%, which is less promising than the earlier results, obtained in incompressible conditions [38]. This matter of fact is not surprising: modeling a gas–liquid mixture leads to many more iterations at both levels. The interesting point is that for severe conditions—$h_{min} = 0.05\,\mu m$, $Sq = 0.1\,\mu m$—the computing time is a third of the deterministic one, with nonetheless a rather good accuracy.

Concerning the curves, the evolution is not monotonous. Even if the convergence criterion is set to an order of magnitude lower, the results remain unchanged. Therefore, the variations are not explained by the convergence criteria on both levels. In addition, there are no more variations in the maximum of the film pressure $P_{max}$ than on the normal load $N$; it suggests that the variations are not due to the local character of a result. By now, we do not have a satisfactory explanation for the load and the pressure shapes. As for the computing time, things are different. Depending on the pressure set at the corners of a macro-element, more or less iterations are needed: if the corner pressure values differ largely from a top-scale iteration to the next, a subsequent number of iterations will occur at the bottom scale.

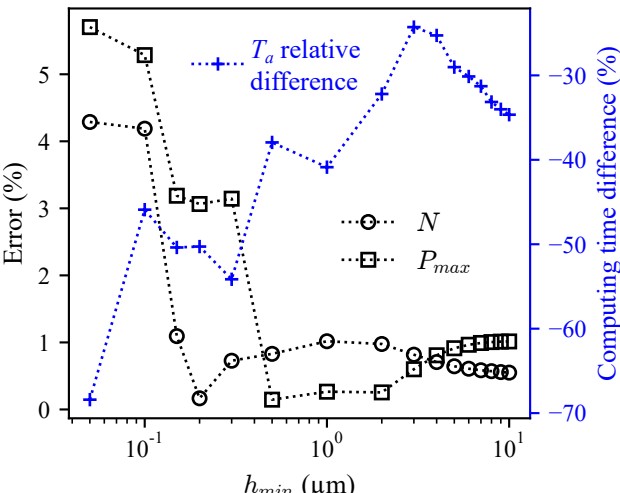

**Figure 12.** Minimum of the film thickness $h_{min}$ influence on the performance of the FMS model. The results are the differences between the FMS and the deterministic models. The left scale is used for the results of relative differences between the models, and the right scale is used for the apparent computing time $T_a$ relative differences.

Compared to the deterministic solution, the FMS approach qualitatively exhibits the same pressure field. As shown in Figure 13, and despite a 6% $P_{max}$ difference, the pressure fields are very close to each other.

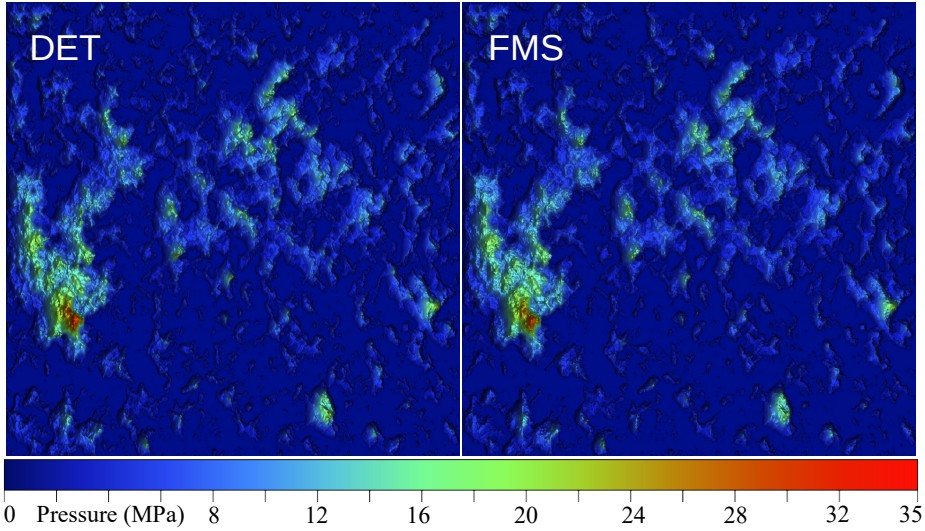

**Figure 13.** Pressure fields: deterministic model (DET) and full multiscale (FMS). $h_{min} = 0.05\,\mu\text{m}$.

As for the residual maps, Figure 14, it is interesting to remark that the trends are qualitatively the same: the higher residuals zones roughly match. Note the grid on the FMS picture: the residuals are set to null on the lines, except the intersection nodes. Indeed, the intersection nodes are the top-scale nodes, but elsewhere on the grid, the nodes are boundary conditions for the inner macro-element nodes. It is important to keep in mind that the convergence criterion, whilst being of the same kind—as often, based on $\frac{\delta P}{P}$ values—cannot be thoroughly compared: in the deterministic model, the whole pressure field is updated at once, whereas in the FMS model, it is updated by parts and, more specifically, for each macro-element. That explains why the residuals are qualitatively compared.

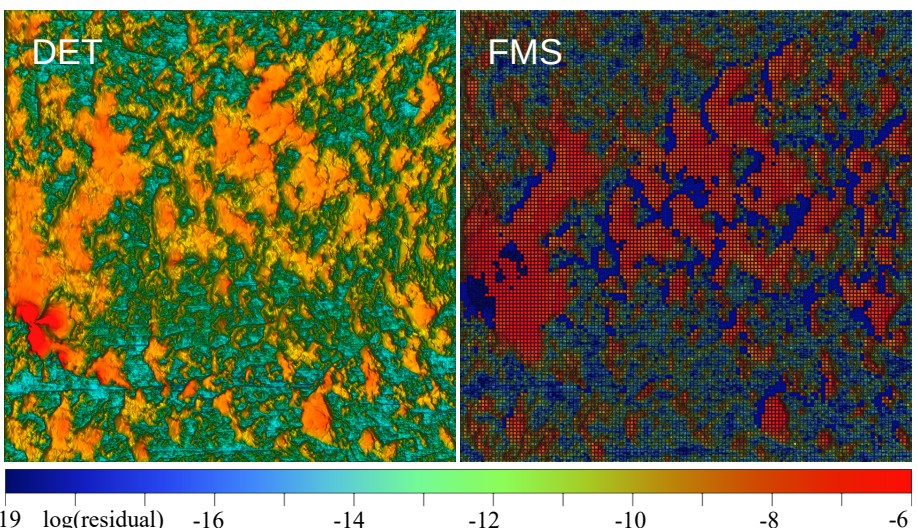

**Figure 14.** Residual field: deterministic model (DET) and full multiscale (FMS). $h_{min} = 0.05\,\mu m$.

### 3.3. Beyond FMS: Hybrid Multiscale (HMS)

When dealing with the HMS—Hybrid Multiscale—the macro-elements are filtered according to the related film thickness. The basic idea is to only refine the macro-elements for which the roughness may influence the lubrication. Simply put, a film thickness threshold $h_T$ is set: below $h_T$, the macro-elements are refined as in the FMS, whereas each macro-element above $h_T$ is treated as a single finite element. The film thickness threshold is deduced from the percentage of the bottom-scale elements set by the user.

Figure 15 is a part of the parametric study that aims to guide the user regarding the percentage of the BS elements to set as a function of the minimum of the film thickness $h_min$. The whole study can be found in SM, Section S3.

Unfortunately, there is no clear strategy rising from the study. To help the understanding of the plots in Figure 15, in the light green areas, the error ($N$ and $P_{max}$) is below 10%, whereas in the dark green areas, the error is below 5%. In both cases, the computing time is at least half the deterministic computing time. When the four configurations are compared—$h_{min} = 0.05, 0.15, 1.00,$ and $5.00\,\mu m$—it is hard to give any advice on the choice of the percentage of the BS elements to set. Surprisingly, in the harsher case, no BS element is needed to reach very satisfactory results, whereas 100% is necessary for the triple of $h_{min}$. So far, different tracks were followed in vain to solve this paradox, and the next explores deals with the BS criterion based on the average film thickness. The parametric study contains other values of $h_{min}$; see SM Section 3 for the complete study.

Anyway, if approximated results are sought—up to a 10% error on $P_{max}$ or $N$—then the HMS is the right tool because of its fastness (a tenth of the deterministic computing time). As an example, the case $h_{min} = 0.05\,\mu m$ is handled about ten times faster with half BS elements. The counterpart is obviously the precision; however, it remains below 10%. Figure 16 shows that the pressure field is in good agreement with the deterministic case. Moreover, the convergence residuals are uniformly spread, like the BS elements in Figure 17, which suggests that the pressure field is a fairly good approximation everywhere.

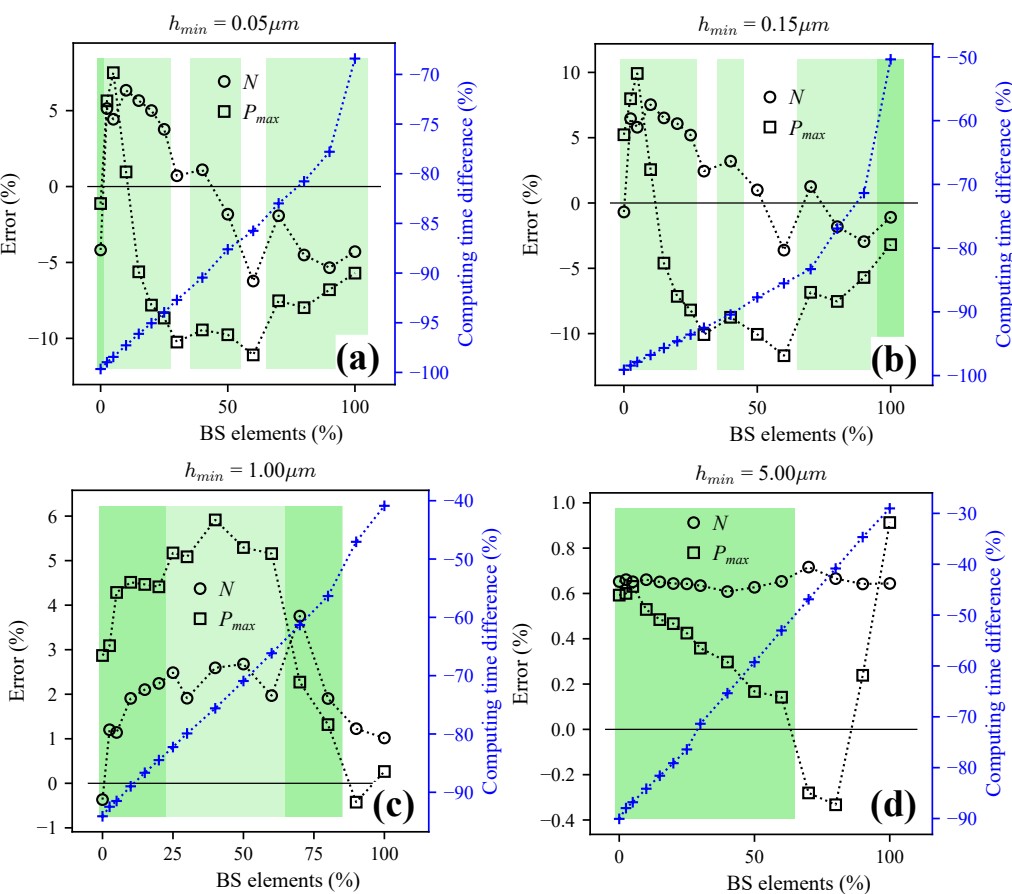

**Figure 15.** Influence of the percentage of bottom scale (BS) on $N$, $P_{max}$, and $T_r$. (**a**) $h_{min} = 0.05\,\mu m$, (**b**) $h_{min} = 0.15\,\mu m$, (**c**) $h_{min} = 1.00\,\mu m$, (**d**) $h_{min} = 5.00\,\mu m$. The values are normalized with respect to the deterministic case. The green areas correspond to errors below 5% and a computing time at least half the deterministic one. The light green areas correspond to errors below 10% and a computing time at least half the deterministic one.

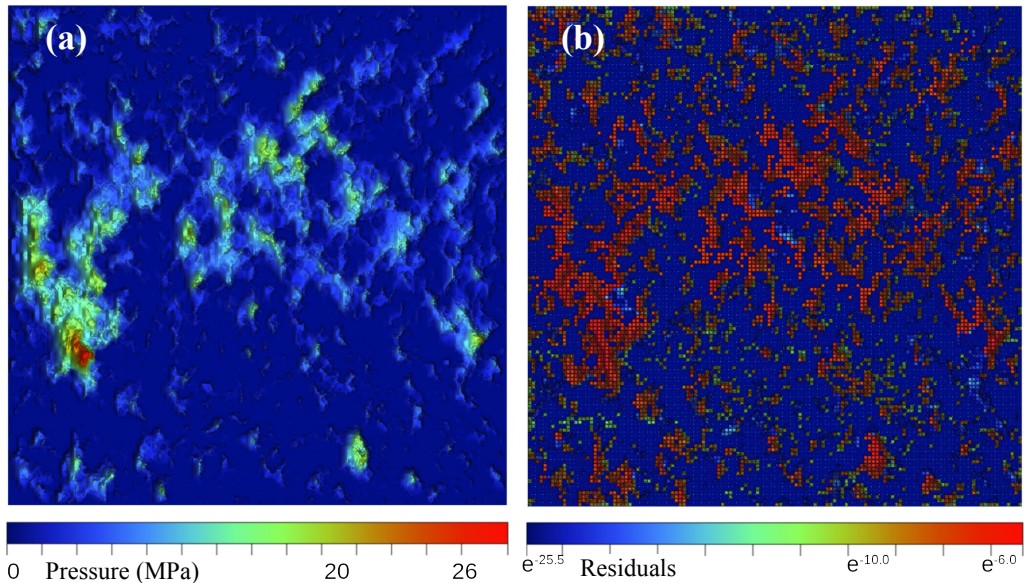

**Figure 16.** Pressure (**a**) and residual (**b**) fields: HMS model, 50% BS-elements, $h_{min} = 0.05\,\mu m$.

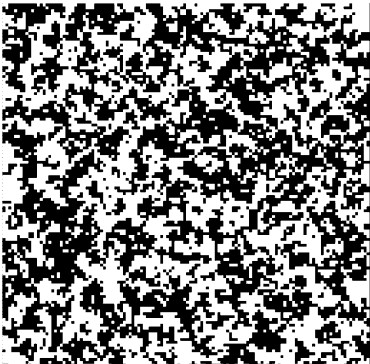

**Figure 17.** BS-elements distribution, 50% BS elements, $h_{min} = 0.05\,\mu m$. The black squares are the BS elements.

The FMS, and even more the HMS, provides for the user with fast approximations of the pressure field. However, a natural question arises: what about simply reducing the mesh size in the deterministic case?

*3.4. Mesh Size Reduction*

Four meshes are used to assess the mesh influence on $P_{max}$, $h_{min}$, and the relative apparent computing time differences: the $513 \times 513$ nodes, $641 \times 641$ nodes, $769 \times 769$ nodes, and $897 \times 897$ nodes. The four models are deterministic. The recurring question about coarsening is, what kind of interpolation would be best? Unfortunately, there is no good answer because not all the information from a fine grid can be transferred to coarser grids. Specifically, the roughness height extrema are likely to disappear during the coarsening step. Therefore, between the three interpolation methods that were tested—namely the approximation, linear, and cubic spline—the linear method is chosen because it is simply the most common. The reader is referred to SM, Figure S8, for being convinced that the linear interpolation remains a good compromise.

As shown in Figure 18, when using the coarsest mesh $513 \times 513$, the computing time falls to values of the same order as those for the HMS. The spikes located at $h_{min} = 0.15\,\mu m; 3\,\mu m$ are explained by slightly higher computing times that are emphasized by the relative differences calculus. The error on $P_{max}$ and $h_{min}$ is even better than for the HMS, Figures 19 and 20.

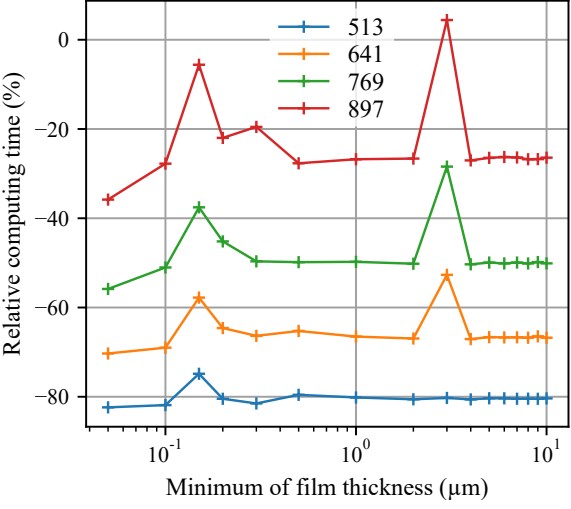

**Figure 18.** Computing time reduction as a function of $h_{min}$ for different meshes.

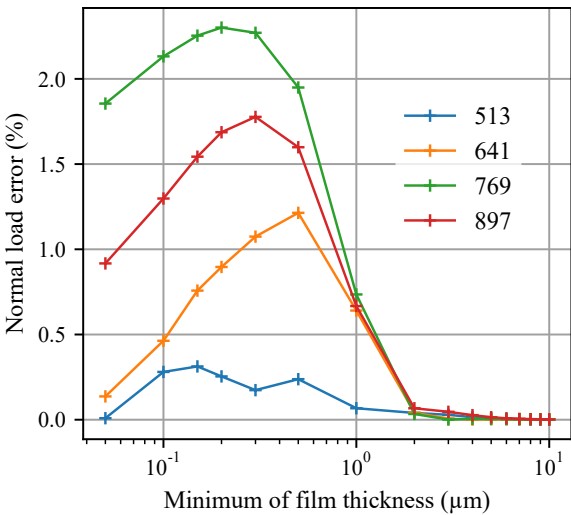

**Figure 19.** Load error as a function of $h_{min}$ for different meshes.

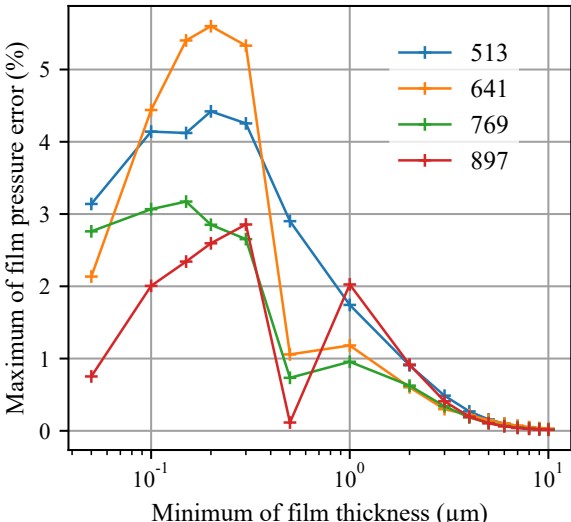

**Figure 20.** Pressure error as a function of $h_{min}$ for different meshes.

If Figure 13—DET—and Figure 21 are compared, it appears that indeed the results are qualitatively very close. The explanation is as follows. With six elements per autocorrelation length—a roughly 513 × 513 nodes mesh, because $Rcl \approx 80$—the mean shape of the surface is well caught. Yet, the main contributors to the lubrication are long wavelengths, which explains that finer meshes bring few additional details.

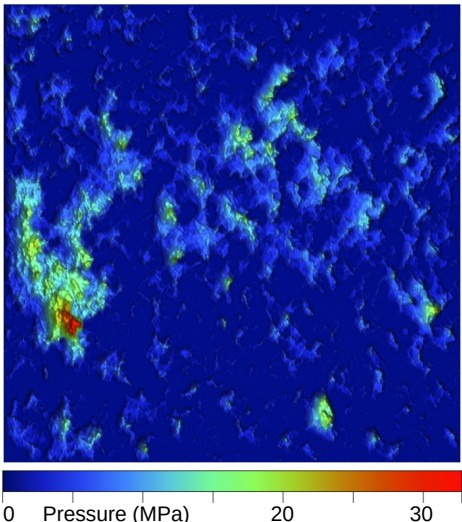

**Figure 21.** Pressure field: deterministic model, $513 \times 513$, $h_{min} = 0.05\,\mu m$.

## 4. Conclusions

In the incompressible case, the FMS model provides for a very fast tool to solve the Reynolds equation. However, strong departures from linearity affect the FMS/HMS model efficiency for compressible fluids. Indeed, with the cavitation phenomenon, the computing process iterates much more: iterations are not only needed at the top-scale level but also at the bottom-scale level. In addition, some relaxation coefficients must be tuned on both levels, and particularly when the ratio $\sigma/h_{min}$ exceeds 1.

Another a priori negative point is that reducing the mesh density allows for the same computing time reduction along with the same result accuracy. And yet, the slider is flat, meaning that the pressure build-up is only generated by the roughness ($Sq = 0.1\,\mu m$). Despite the preceding points, it is worth noting that the computing times—when the HMS is used with a few BS elements—remain attractive, compared to the deterministic model, whether it is reduced or not.

The benefits of the FMS/HMS approach are much more expected in the following cases:

1. Shorter wavelengths—$Rcl > 100$ and $L_{x,y} < 10mm$;
2. Rough texturized surfaces—square dimples being modeled with simple FEM macro-elements, and rough contacting parts discretized with BS elements;
3. As many threads as there are macro-elements—which leads to a GPU implementation of the numerical code.

As for the numerical aspects, additional computing time can be saved.

1. The TS element boundaries are updated once the whole TS element batch is processed. However, some TS elements converge slowly—mainly because of a local narrow slider gap. Therefore, a new criterion has to be set up to locally guarantee the best compromise accuracy/iteration number.
2. The TS element boundary update must be monitored because important changes in the TS pressure field affect the four other connected macro-elements—in particular, oscillations are undesirable.
3. The heights of the element boundaries are the result of the domain division, and this can lead to rough relief for some of them, with convergence problems. In a future work, a sensitivity analysis will study the effect of numerically smoothed boundaries on the slider lubrication. We are confident that the results will not change much and that the Reynolds equation will be solved faster on TS elements.

To conclude the present work, the FMS/HMS approach is accurate and fast due to a highly parallelizable structure, and promising keys to improvement make this technique a good candidate for the lubrication of rough parallel surfaces.

**Supplementary Materials:** The following supporting information can be downloaded at: https://www.mdpi.com/article/10.3390/lubricants10120329/s1, Section S1: The SU-PG method; Subsection S2.1: The weak formulation using the Bubnov-Galerkin method; Subsection S2.2: Couette term upwind weighting; Subsection S2.3: The Newton-Raphson scheme; Section S3: HMS parametric study for the flat rough slider; Figure S8: Influence of the interpolation type on the coarse mesh results related to the fine mesh.

**Author Contributions:** Conceptualization, N.B. and A.F.; methodology, N.B. and A.F.; software, N.B. and A.F.; validation, N.B. and A.F.; writing—original draft preparation, A.F.; writing—review and editing, N.B. and A.F.. All authors have read and agreed to the published version of the manuscript.

**Funding:** This work pertains to the French government program Investissements d'Avenir (LABEX INTERACTIFS, reference ANR-11-LABX-0017-01, and EUR INTREE, reference ANR-18-EURE-0010).

**Data Availability Statement:** Not applicable

**Conflicts of Interest:** The authors declare no conflict of interest.

## Abbreviations

The following abbreviations are used in this manuscript:

| | |
|---|---|
| BS | Bottom Scale |
| FEM | Finite Element Method |
| FMS | Full Multiscale |
| HMS | Hybrid Multiscale |
| LGM | Lubricant General Model |
| MIX | MIXture, same as GLM |
| TS | Top Scale; TS element = a macro-element. |

## Nomenclature

Reynolds equation

| | |
|---|---|
| $e$ | $e^{th}$ finite element |
| $h$ | film thickness |
| $p$ | pressure |
| $l_e$ | streamline length of an element $e$ |
| $N$ | slider normal load |
| $N_i$ | shape and weighting function at node $i$ |
| $P_c$ | cavitation pressure |
| $R_i$ | Reynolds equation residual at node $i$ |
| $\boldsymbol{u}$ | slider speed vector |
| $[B_e]$ | $e^{th}$ element right-hand side |
| $[K_e]$ | $e^{th}$ element elementary matrix |
| $\delta p_j$ | pressure increment at node $j$ |
| $\mu$ | fluid dynamic viscosity |
| $\Omega$ | lubricated contact domain |
| $\rho$ | fluid density |

Penalty method

| | |
|---|---|
| $\kappa$ | arbitrary chosen coefficient for the penalty method |
| $\tau$ | penalty coefficient, $\tau = 10^k$ |

Elrod algorithm

| | |
|---|---|
| $\theta$ | fluid volume fraction |

General model
$p$         partial pressure
$r_g$        the specific gas constant; for dry air $r_g = 287.0 \, \text{J} \, \text{kg}^{-1} \, \text{K}^{-1}$
$C_a$        liquid concentration of a species $a$
$H_a^{cp}$       Henry's constant of a species $a$
$P_{atm}$       atmospheric pressure, $P_{atm} = 1.01 \times 10^5 \, \text{Pa}$
$\lambda$         mass fraction
Multiscale
$h_T$        film thickness threshold
$n$         $= ne_x = ne_y$, total number of BS nodes
$n_e^b$        number of bottom-scale (BS) elements of a TS element, along $x$ or $y$
$n_e^t$        number of top-scale (TS) elements, or macro-elements, along $x$ or $y$
BS        Bottom scale
P         TS pressure
Q         TS nodal mass flow
TS        Top scale
$\{b\}$        subdivided macro-element FEM right-hand side
$[k]$        subdivided macro-element FEM system matrix

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
