# Peer review of "Full and Hybrid Multiscale Lubrication Modeling"

_lubricants, doi:10.3390/lubricants10120329_

Round 1

Reviewer 1 Report

in chapter 4, a few methods are introduced to solve Reynolds equation with cavitation.  however it is not clear later when the results are presented which method was used.  making a more explicit will help teh readers.

I attached the manuscrpit with a few places marked.

Author Response

Thank you for the review.

  • Everywhere ".. the Figure ..." appeared, it is replaced by "Figure ..."
  • idea -> ideas, line 277

We have highlighted the part of the text where the lubricant model used is presented (lines 57 and 162)

Reviewer 2 Report

The author's has a good ideas but the write up was not properly presented. The author's should do the following before this manuscript can be considered.

1. Remove Nomenclature and put it either before reference or after Abstract. And it should not be labeled.

2. Merge section 3 to 5 together because they are gotten from literature, is not your main simulation work.

3. In your method and materials kindly show the mathematical expression behind your study

4. The caption for Figures are too long summarize them and send the explanation to the body of the work.

5. Figure 13 to 16. When having more than one graphs in a figure, there is need to differentiate them with either (a) ND (b). And this should reflect in the caption labels

6. This paper didn't have conclusion section. The discussion is not conclusion. The author's need to add a good conclusion in this manuscript.

7. All the references are out dated, that is they are old once. This doesn't tell well for a good manuscript. The author's should do more work to improve both the methodology and discussion. With that they will be able to get recent article online at least from 2022 to 2020 not less down 10 reference, recent onces should be added, it help to improve it.

After this corrections the author's should submit back for evaluation 

Author Response

Thank you for the review.

Reviewer 3 Report

In this work, the authors present a thorough investigation on full and hybrid multiscale lubrication modeling. The analyzes are convincing, the conclusions are well thought out and systematically presented. So, this paper can be published after minor revision.
In the introduction section, the author describes the research background of the article in detail, but there are few references in the past three years, which cannot reflect the current innovation and need to be updated. If the second part Nomenclature, the third part Equations, and the fourth part the cavitations are necessary to separate into segments, can they merge?

Author Response

Thank you for the review.

We agree that the paper lacks in recent studies. We have enriched the cavitation and the multiscale background accordingly.

Round 2

Reviewer 2 Report

The authors has done major of the work I requested they do. However, the authors need to work on the Conclusion with the following point

1. Remove all citations from the conclusion section

2.  instead of using a bullet, it is better to use a number numeral

Author Response

Thank you for the comment, we appreciate your approval.

  1. Remove all citations from the conclusion section - done
  2. instead of using a bullet, it is better to use a number numeral - done